

# The mitochondrial phylogeny of land plants shows support for Setaphyta under composition-heterogeneous substitution models

Filipe Sousa[1], Peter Civáň[1,2], João Brazão[1], Peter G. Foster[3] and Cymon J. Cox[1]

[1] Centro de Ciências do Mar, Universidade do Algarve, Faro, Portugal
[2] INRAE-Université Clermont-Auvergne, Clermont-Ferrand, France
[3] Department of Life Sciences, Natural History Museum, London, United Kingdom

## ABSTRACT

Congruence among analyses of plant genomic data partitions (nuclear, chloroplast and mitochondrial) is a strong indicator of accuracy in plant molecular phylogenetics. Recent analyses of both nuclear and chloroplast genome data of land plants (embryophytes) have, controversially, been shown to support monophyly of both bryophytes (mosses, liverworts, and hornworts) and tracheophytes (lycopods, ferns, and seed plants), with mosses and liverworts forming the clade Setaphyta. However, relationships inferred from mitochondria are incongruent with these results, and typically indicate paraphyly of bryophytes with liverworts alone resolved as the earliest-branching land plant group. Here, we reconstruct the mitochondrial land plant phylogeny from a newly compiled data set. When among-lineage composition heterogeneity is accounted for in analyses of codon-degenerate nucleotide and amino acid data, the clade Setaphyta is recovered with high support, and hornworts are supported as the earliest-branching lineage of land plants. These new mitochondrial analyses demonstrate partial congruence with current hypotheses based on nuclear and chloroplast genome data, and provide further incentive for revision of how plants arose on land.

## INTRODUCTION

The embryophytes, or land plants, share a green algal ancestor (*McCourt, Delwiche & Karol, 2004*) that colonized terrestrial environments between 515.1–470.0 Ma (*Morris et al., 2018*) and comprise gametophyte-dominant lineages, collectively known as bryophytes, and a sporophyte-dominant lineage, the tracheophytes. Establishing the phylogenetic relationships between bryophytes (mosses, liverworts and hornworts) and tracheophytes (a monophyletic lineage that includes lycopods, ferns and seed plants) is therefore fundamental to understanding the evolution of plants on land. However, phylogenetic inferences of relationships among embryophytes drawn from molecular data of the nuclear (*Finet et al., 2010*; *Wodniok et al., 2011*; *Wickett et al., 2014*), chloroplast (*Cox et al.,*

Corresponding author
Cymon J. Cox, cymon@ualg.pt

*2014*; *Ruhfel et al., 2014*; *Zhong et al., 2013*; *Bell et al., 2020*), and mitochondrial (*Turmel, Otis & Lemieux, 2013*; *Liu et al., 2014*; *Bell et al., 2020*) genomes have long remained conflicting. These incongruences are likely due to molecular evolutionary processes that are especially apparent at deep timescales, such as multiple substitutions on the same site, that lead to loss of phylogenetic signal, and heterogeneity in substitution process patterns among sites and among lineages (*Cox, 2018*). Phylogenetic patterns commonly observed among embryophytes included a sister-group relationship between hornworts and other embryophytes (*Hedderson, Chapman & Rootes, 1996*; *Malek et al., 1996*; *Nishiyama & Kato, 1999*; *Wickett et al., 2014*), between liverworts and other embryophytes (*Lewis, Mishler & Vilgalys, 1997*; *Karol et al., 2001*; *Qiu et al., 2006*; *Gao, Su & Wang, 2010*; *Karol et al., 2010*; *Clarke, Warnock & Donoghue, 2011*), or between embryophytes and a clade uniting mosses and liverworts (*Karol et al., 2010*). An alternative pattern shows an initial split between the bryophytes and the tracheophytes (*Hori, Lim & Osawa, 1985*; *Nishiyama et al., 2004*; *Goremykin & Hellwig, 2005*; *Cox et al., 2014*; *Wickett et al., 2014*; *Puttick et al., 2018*; *Sousa et al., 2019*; *Leebens-Mack et al., 2019*), implying the monophyly of both lineages. Nevertheless, the results are very much dependent on data and methodology, with authors typically presenting competing hypotheses. For instance, several recent phylogenomic analyses based on large nuclear data sets and extensive taxon sampling (e.g., *Wickett et al., 2014*; *Leebens-Mack et al., 2019*) have been equivocal. These studies presented monophyletic-bryophyte phylogenies based on multi-species coalescent supertrees, but concatenated analyses of the same data resulted in trees in which the bryophytes were paraphyletic. Consequently, the authors were unable to provide arguments for which hypothesis was to be preferred. Indeed, the efficacy and suitability of using multi-species coalescent summary analyses versus concatenated data analyses for phylogenies with deep timescales is currently a topic of considerable debate (e.g., *Tonini et al., 2015*; *Edwards et al., 2016*; *Springer & Gatesy, 2016*). However, it should be noted that concatenated analyses of nuclear data do support a monophyletic bryophyte clade when modeling composition heterogeneity across the tree, although restricted analytical conditions, namely reduced taxon and site sampling, are currently necessary to decrease computational complexity when using tree-heterogeneous composition models. For instance, to use these models, *Sousa et al. (2019)* analysed a reduced amino acid data set of 26 taxa and 100 genes, while *Puttick et al. (2018)* analysed Dayhoff-recoded data, that reduces the amino acid data to only six character states. Nevertheless, these tree-heterogeneous composition models are demonstrably better-fitting and are therefore likely more accurate and reliable than the analyses of larger data sets that used simpler and poorer-fitting models (*Cox, 2018*).

Most analyses of land plant relationships have been based on chloroplast data and have typically shown the bryophytes to be paraphyletic (see Table 1 in *Cox, 2018*). More recent phylogenetic analyses using models that account for saturation and composition tree-heterogeneity have, however, indicated that the bryophytes form a monophyletic group, and the authors provided arguments as to why these analyses using better-fitting models are to be preferred (*Cox et al., 2014*). In contrast, few land plant analyses of mitochondrial data have been presented, but all have indicated that the bryophytes form a paraphyletic group (i.e., *Duff & Nickrent, 1999*; *Groth-Malonek et al., 2004*; *Liu et al., 2014*).

The most recent and extensive phylogenetic analyses of plant mitochondrial genomes using tree-homogeneous composition models have shown that protein-coding nucleotide data place mosses as the sister-group to the remaining embryophytes, whereas amino acid data show a split between liverworts and the remaining embryophytes (*Liu et al., 2014*). However, this placement of liverworts as the sister-group to the remaining embryophytes contradicts recent nuclear and chloroplast phylogenies which show high support for the clade Setaphyta, that groups liverworts with mosses (*Cox et al., 2014*; *Puttick et al., 2018*; *Sousa et al., 2019*).

Our confidence in any evolutionary hypotheses regarding land plant relationships would increase greatly if phylogenetic inferences made from all three plant genomic compartments were not in conflict. In this study we investigate the mitochondrial phylogeny of land plants by applying better-fitting evolutionary models that account for composition tree-heterogeneity to a newly compiled data set of mitochondrial loci that includes sequences from three newly assembled genomes from the Coleochaetales and Zygnematales. We assemble a mitochondrial land plant data set of 26 taxa, which includes all major lineages of land plants and one of the putative most closely-related lineages to land plants, the Zygnematales. Assuming that both bryophytes and tracheophytes are likely monophyletic on the species tree (*Sousa et al., 2019*), the taxon sampling is deliberately restricted to include a proportional selection of bryophytes (11 taxa) and tracheophytes (10) as symmetrical trees improve estimation (*Huang & Knowles, 2009*) and minimise long branch attraction (*Philippe & Laurent, 1998*). Where possible, taxa were chosen to span what is currently considered the ancestral node of each bryophyte lineage (or the oldest ancestral node possible given the currently available data), thereby attempting to minimise the length of the sub-tending branch of each bryophyte clade and reduce long-branches. Most importantly, a smaller data set enables us to use better-fitting models that account for among-lineage and among-site composition heterogeneity that are computationally intractable for large data sets. This reduced sampling is in contrast to other studies which typically include many more taxa from highly derived lineages (especially angiosperms) whose inclusion, we maintain, has little impact on the resolution among major lineages (the question under consideration), but severely restricts the complexity of models that can be used and therefore the reliability of the inferences. Notably, a recent large-scale analysis of plant transcriptomes, despite the inclusion of 1,155 taxa and 410 genes, was unable to resolve many of the long-standing contentious phylogenetic relationships, such as the relationships among the major lineages of land plants (Fig. 3 in *Leebens-Mack et al., 2019*). Indeed, rather than just including all available data injudiciously, and thereby restrict the complexity of models that can be applied for phylogenetic inference, it is important to consider which data should be included in a particular analysis: the choice of data should reflect carefully the question being addressed and its suitability for analysis by better-fitting models of molecular evolution.

## MATERIALS & METHODS

### Sampling of mitochondrial genomes

We sampled 21 taxa representing the major lineages of land plants, plus 5 green algae species as outgroup taxa. The taxa sampled for this study were: Coleochaetales (*Chaetosphaeridium globosum*, *Coleochaete scutata*), Zygnematales (*Closterium baillyanum*, *Gonatozygon brebissonii*, *Roya anglica)*, liverworts (*Aneura pinguis. Marchantia polymorpha*, *Pleurozia purpurea*, *Treubia lacunosa*), mosses (*Atrichum angustatum*, *Bartramia pomiformis*, *Physcomitrella patens*, *Sphagnum palustre*, *Tetraphis pellucida*), hornworts (*Megaceros aenigmaticus*, *Phaeoceros laevis*), lycophytes (*Huperzia squarrosa*, *Isoetes engelmannii*), pteridophytes (*Ophioglossum californicum*, *Psilotum nudum*), and spermatophytes (*Brassica napus*, *Cycas taitungensis*, *Ginkgo biloba*, *Liriodendron tulipifera*, *Oryza sativa*, *Welwitschia mirabilis*).

Algal cultures for *Coleochaete scutata* (SAG 3.91) and *Gonatozygon brebissonii* (SAG 292) were obtained from the Culture Collection of Algae (SAG) (Georg-August-Universität Göttingen, Germany), and the algal culture of *Roya anglica* (ACOI 799) was obtained from the Coimbra Collection of Algae (ACOI) (Universidade do Coimbra, Coimbra, Portugal). The mitochondrial genomes of the three taxa were sequenced and assembled *de-novo* using standard methods as described in *Civán et al. (2014)*, and annotated with the aid of Mitofy (*Alverson et al., 2010*) and NCBI BLAST analyses (*Altschul et al., 1990*). The remainder of the genomes were retrieved from NCBI GenBank (Bethesda, USA). The list of species samples, their classification, and the source and accession numbers of the sequences used are shown in Table 1.

### Alignment and model testing

The sequences of each of 43 mitochondrial protein-coding genes were aligned using the program MAFFT (vers. 6.8; *Katoh & Toh, 2008*). Nucleotide alignments were manually edited in Geneious (vers. 9; http://www.geneious.com) to remove out-of-frame indels, misaligned portions, premature stop codons, positions with less than 50% taxon occupancy, and ambiguities. Genes that were missing within the algal outgroup taxa (Coleochaetales and Zygnematales) or were under 200 bp in length were discarded. The final data set consisted of 36 genes for 26 taxa, with 9.5% of gene sequences missing from the alignment. Amino acid alignments were generated by translation from each of the 36 nucleotide matrices using SeaView (vers. 4.5.4; *Gouy, Guindon & Gascuel, 2009*). The best-fitting substitution models for the 36 amino acid alignments were inferred in PartitionFinder (*Lanfear et al., 2012*) using the BIC selection criterion. The stmtREV (*Liu et al., 2014*) with a gamma-distribution of among-site rates (+G) model was the best-fitting model for all genes but one, for which the chosen model was JTT+G (*Jones, Taylor & Thornton, 1992*).

### Gene tree estimation and monophyly test

Gene trees were estimated from individual nucleotide matrices, using the general time-reversible model (*Rodriguez et al., 1990*) with a gamma rate distribution and estimated base frequencies (GTR+G+Fest), and from amino acid matrices, using the best-fitting models inferred inPartitionFinder (stmtREV+G+$F_{est}$, JTT+G+$F_{est}$). Bayesian MCMC analyses

**Table 1  Accession table of the 26 samples used in this study.** For each species, the corresponding taxonomic group and NCBI GenBank accession numbers are shown. Accessions marked with an asterisk (*) correspond to newly assembled genomes.

| Species | Taxonomic group | Accession |
|---|---|---|
| *Closterium baillyanum* | Zygnematales | NC_022860 |
| *Gonatozygon brebissonii* | Zygnematales | MK720950* |
| *Roya anglica* | Zygnematales | MK720948* |
| *Chaetosphaeridium globosum* | Coleochaetales | NC_004118 |
| *Coleochaete scutata* | Coleochaetales | MK720949* |
| *Megaceros aenigmaticus* | hornworts | NC_012651 |
| *Phaeoceros laevis* | hornworts | NC_013765 |
| *Aneura pinguis* | liverworts | NC_026901 |
| *Marchantia polymorpha* | liverworts | NC_001660 |
| *Pleurozia purpurea* | liverworts | NC_013444 |
| *Treubia lacunosa* | liverworts | NC_016122 |
| *Atrichum angustatum* | mosses | NC_024520 |
| *Bartramia pomiformis* | mosses | NC_024519 |
| *Physcomitrella patens* | mosses | NC_007945 |
| *Sphagnum palustre* | mosses | NC_024521 |
| *Tetraphis pellucida* | mosses | NC_024290 |
| *Welwitschia mirabilis* | seed plants | NC_029130 |
| *Brassica napus* | seed plants | NC_008285 |
| *Liriodendron tulipifera* | seed plants | NC_021152 |
| *Oryza sativa* | seed plants | NC_011033 |
| *Cycas taitungensis* | seed plants | NC_010303 |
| *Ginkgo biloba* | seed plants | NC_027976 |
| *Ophioglossum californicum* | ferns | NC_030900 |
| *Psilotum nudum* | ferns | NC_030952, KX171639 |
| *Huperzia squarrosa* | lycophytes | NC_017755 |
| *Isoetes engelmannii* | lycophytes | FJ010859, FJ176330, FJ390841, FJ536259, FJ628360 |

were performed with P4 (*Foster, 2004*) using tree-homogeneous (henceforth referred to as CV1, i.e., one composition vector) and tree-heterogeneous (NDCH) composition models. Each analysis had two independent runs which were assessed for convergence by calculation of the marginal likelihood (chains were considered to have converged if they differed by <10 units) and of the average standard-deviation of split support between trees sampled from the posterior distributions (chains were considered to have converged if <0.01). Posterior predictive simulations of the $X^2$ test statistic of composition homogeneity was used to assess composition fit (*Foster, 2004*). For each gene, 50% majority-rule consensus trees were generated from the best-fitting analyses of the nucleotide and amino acid data.

Gene tree topologies inferred from each of the 36 amino acid alignments were tested using "gene genealogy interrogation" (GGI; *Arcila et al., 2017*) to ascertain whether the non-monophyly of the five major lineages under scrutiny (hornworts, liverworts, mosses, tracheophytes, and the outgroups) was statistically supported by the data. We
were to consider any gene that supported the non-monophyly of one of these clades as aberrant and not suitable for inclusion in the combined analyses, as the monophyly of hornworts, liverworts, mosses, tracheophytes and embryophytes has been consistently recovered in molecular phylogenies (e.g., *Qiu et al., 2006*; *Liu et al., 2014*). Optimal trees for each gene were compared to each of fifteen constraint trees representing all possible topologically resolved combinations of the five monophyletic groups, using a nonparametric bootstrapping test. The results were assessed for statistical significance with the Approximately Unbiased (AU) test (*Shimodaira, 2002*) in CONSEL (vers. 0.1k; *Shimodaira & Hasegawa, 2001*). Optimal trees were estimated in RAxML (MPI-compiled vers. 8.2.8; *Stamatakis, 2014*) using the model stmtREV+G+$F_{est}$ and starting from 20 random trees. Constraint trees were written in Newick format with internal branches within each of the five major clades collapsed to a polytomy. Each constraint tree was optimized in RAxML under the stmtREV+G+$F_{est}$ model. Constraint topologies were considered statistically supported by the data if the *p*-value of the AU test was equal or greater than 0.05, meaning that the monophyly of each of the five lineages could not be rejected. In every gene, at least one of the constraint topologies was supported by the data, meaning that the monophyly of each clade could not be rejected, thus all 36 genes were included in downstream analyses.

## Analyses of concatenated nucleotide data

A nucleotide alignment with 24,864 characters was obtained from the concatenation of the 36 individual genes. A second concatenated nucleotide matrix was constructed by codon-degenerate recoding of the data where ambiguity codes are used to negate synonymous substitutions (*Criscuolo & Gribaldo, 2010*; *Cox et al., 2014*). Bayesian MCMC analyses were performed on the concatenated and codon-degenerate nucleotide matrices using a tree-homogeneous composition model (CV1; 2 replicates) and the tree-heterogeneous composition (NDCH2; 4 replicates) model, as implemented on P4. In contrast to the original NDCH model (*Foster, 2004*) that allows an *a priori* defined number of compositions to evolve on the tree, the NDCH2 model estimates a separate composition for each node of the tree, constrained by a sampled Dirichlet prior on how much the composition vectors may differ from the empirical composition. Model fit to composition heterogeneity was inferred during the Bayesian MCMC with posterior predictive simulations of the $X^2$ statistic of composition homogeneity, where *p*-values equal or greater than 0.05 indicate acceptance of the model. The GTR+G+$F_{est}$ model of substitution was used for for all MCMC of nucleotide and degenerate nucleotide data. Marginal likelihoods were estimated in P4 according to the eq16 method of *Newton & Raftery (1994)*.

## Analyses of concatenated amino acid data

An amino acid alignment with 8,288 characters was obtained by concatenation of the amino acid translations of the 36 genes. Bayesian MCMC analysis was performed on the concatenated amino acid data using both tree-homogeneous composition (stmtREV+G+$F_{CV1}$; 2 replicates) and tree-heterogeneous composition (stmtREV+G+$F_{NDCH2}$; 4 replicates) models, with the fit of the composition evaluated

by posterior predictive simulations as described for the nucleotide data. In addition, a Bayesian MCMC analysis was also performed using PhyloBayes (MPI-compiled vers. 1.6; *Lartillot, Lepage & Blanquart, 2009*) under the model stmtREV+G+$F_{CAT}$.

Alignments of individual genes, the concatenated data, and the resulting tree files of each analysis are available on Zenodo (doi: 10.5281/zenodo.3554149). All ML analyses of the concatenated nucleotide and amino acid data sets were consistent with the homogeneous Bayesian MCMC analyses and are not reported here, but the resulting tree files are also available on Zenodo.

## RESULTS

### Nucleotide and codon-degenerate data

All individual genes were best-fit by a tree-heterogeneous composition model with two composition vectors on the tree (CV2; Table S1). Majority-rule consensus trees resulting from the best-fitting Bayesian MCMC analyses of individual genes had low resolution in general, but liverworts were supported (>95% posterior probability (PP)) as the earliest-branching lineage in two genes (*nad* 3 and *nad* 5), whereas the mosses were supported as the earliest-branching lineage in one gene (*ccm* C). All other resolutions of the bryophyte lineages relative to the tracheophyte clade were not statistically supported, and the Setaphyta clade was not resolved in any gene tree.

Bayesian MCMC analysis of the concatenated nucleotide data set, assuming a homogeneous composition (CV1), resulted in a tree with mosses as the sister-group to the remaining land plants (PP = 1.0; Fig. 1A; Fig. S1), and hornworts as sister-group to the tracheophytes (PP = 1.0). In contrast, the analysis of the degenerate data set under a homogeneous base composition (Fig. 1C; Fig. S2) returned a tree with liverworts the earliest-branching lineage of embryophytes (PP = 1.0) and hornworts the sister-group to tracheophytes (PP = 0.93). However, the homogeneous model was rejected for both data sets by the posterior predictive simulation of the $X^2$ statistics of homogeneity, with a tail-area probability of 0.0, thereby indicating that the data were not composition homogeneous. The tree-heterogeneous composition analysis (NDCH2) of the concatenated nucleotide data also resulted in mosses supported as the earliest-diverging land plant lineage (PP = 0.98; Fig. 1B; Fig. S3), but placed liverworts as the sister-group to tracheophytes (PP = 0.94). The NDCH2 model was a good fit to the data according to the posterior predictive simulation ($X^2$ $p = 0.99$). When analysing the codon-degenerate data, the NDCH2 model recovered hornworts as the sister-group to the remaining land plants with full branch support (PP = 1.0; Fig. 1D; Fig. S4), and mosses fully supported (PP = 1.0) as the sister-group to liverworts (i.e., the clade Setaphyta). The NDCH2 model was not a good statistical fit to the data for the best scoring MCMC run, according to the $X^2$ posterior predictive test ($X^2$ $p = 0.038$; Fig. S4).

### Amino acid data

Individual mitochondrial protein alignments were best-fit by both homogeneous and heterogeneous composition models, with some being best-fit by a model with up to four compositions (CV4), indicating that they are highly heterogeneous in composition among

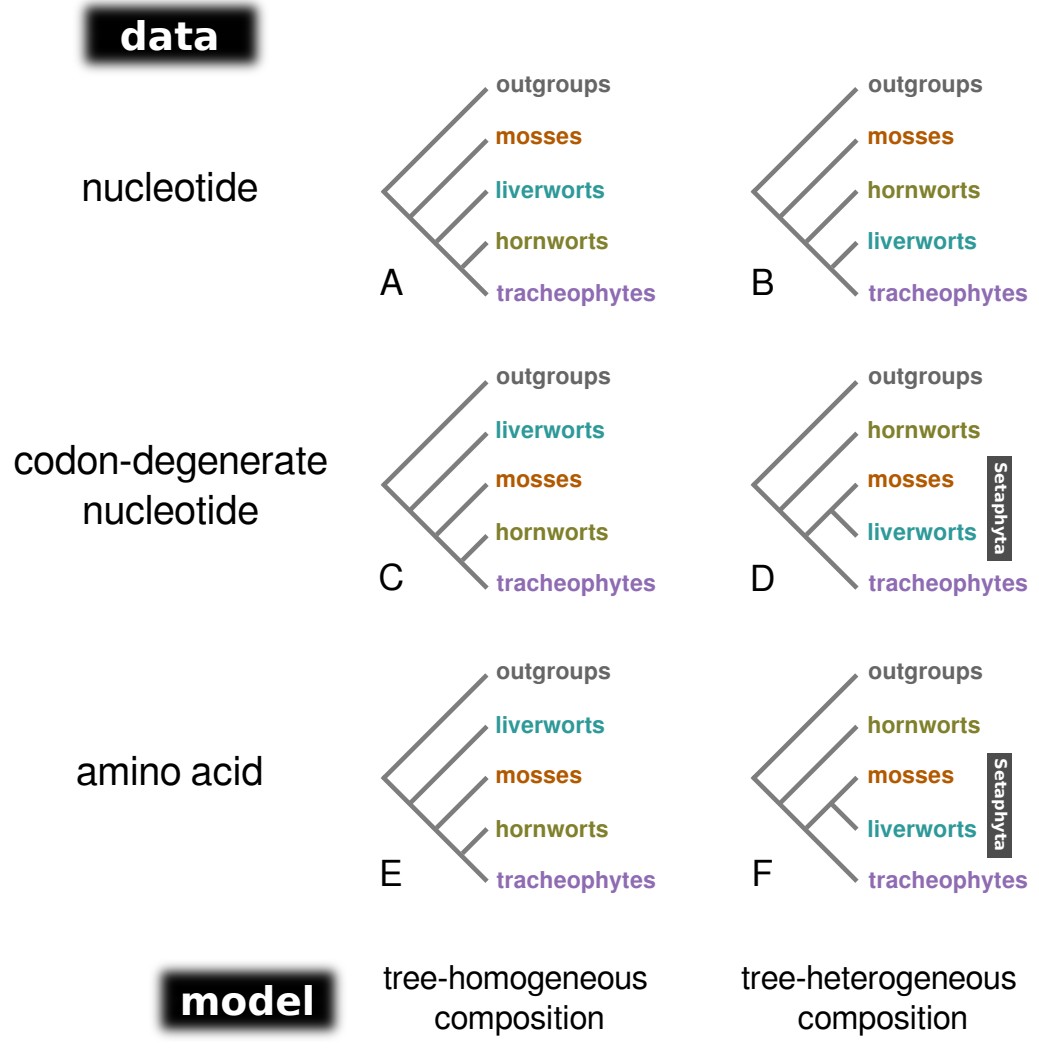

**Figure 1** **A schematic representation of the topologies obtained from tree-homogeneous and tree-heterogeneous analyses of nucleotide, codon-degenerate nucleotide, and amino acid translation data.** Analyses of nucleotide data place mosses as the earliest-branching lineage of the embryophytes (A, B). Analyses of codon-degenerate and amino acid data under tree-homogeneous models place liverworts as the sister-group to the remaining embryophytes (C, E), whereas analyses under tree-heterogeneous models show support for the clade Setaphyta (D, F).

lineages (Table S1). Majority-rule consensus trees of best-fitting Bayesian MCMC analyses of individual proteins were poorly supported regarding relationships among bryophyte lineages. Indeed, only one tree (*ccm* C) showed any statistically supported resolution and placed the mosses as the earliest-branching lineage of embryophytes. Although not statistically supported, one amino-acid tree showed bryophytes as a monophyletic group (*rps* 7), and the clade Setaphyta was present in three others (*atp* 4, *rpl* 2, and *sdh* 4).

When the concatenated amino acid data were analysed with a tree-homogeneous composition model (CV1), the resulting tree recovered liverworts as the sister-group to the remaining embryophytes without statistically significant support (PP = 0.82), and

hornworts as the sister-group to tracheophytes, also without support (PP = 0.82; Fig. 1E; Fig. S5). The homogeneous composition model did not fit the data ($X^2$ $p = 0.0$). The Phylobayes stmtREV+G+$F_{CAT}$ analysis (stationary, data-heterogeneous composition model) of the concatenated amino acid data resulted in an unsupported placement of liverworts as the sister-group to the remaining embryophytes (PP = 0.89; Fig. S6). A posterior predictive composition homogeneity test using Phylobayes (*readpb_mpi* parameter -*comp*) showed that the data rejected the model and that the data were therefore composition tree-heterogeneous ($p = 0.0$). When the tree-heterogeneous composition NDCH2 model was used to model the concatenated protein data, the separate analyses did not converge on the same tree topology, although the NDCH2 model was a good fit ($X^2$ $p = 0.1022$). The tree obtained from the analyses with the best marginal likelihood (-lnLh 142829.1129) supported hornworts as the earliest-branching lineage (PP = 1.0), with the liverworts as the sister-group to mosses (PP = 1.0; Figs. 1F, 2, Fig. S7).

## DISCUSSION

### Gene tree discordance in mitochondrial data is likely due to mis-modeling and insufficient phylogenetic signal

Alternative hypothesis testing using nonparametric and parametric bootstrapping has been applied before to the mitochondrial land plant phylogeny to test the fit of the data (*Liu et al., 2014*). Here we chose a different approach, and used the optimized likelihood of constraint trees to identify genes that did not support the monophyly of the major embryophyte lineages (hornworts, liverworts, mosses, and tracheophytes) and of the outgroup. The optimal trees of the 36 mitochondrial genes, inferred under maximum-likelihood, show varied topologies, among which none is predominant. Because the four major land plant lineages are known to be monophyletic (as shown by many studies, e.g., *Wickett et al., 2014*) our concern was to identify gene trees that showed non-monophyly of one of these groups. The strategy we adopted allowed us to discern whether such topologies truly reflect underlying data or whether they are one among different topologies supported by the data. The AU test indicated, in all genes, that the monophyly of each land plant lineage was not statistically contradicted. This result suggests that the observed phylogenetic conflict among mitochondrial gene trees is unlikely to be explained by biological processes, such as horizontal gene transfer or duplication-loss, affecting specific lineages within each of the four major groups, and that any observed paraphyly of major groups on gene trees is probably the result of inadequate data modeling or paucity of phylogenetic signal.

### Synonymous substitutions are responsible for the placement of mosses as the earliest-branching lineage of embryophytes

The tree inferred from the concatenated nucleotide data set of 36 mitochondrial genes shows mosses as the sister-group to the remaining land plants, as previous analyses of mitochondrial nucleotide data have shown (*Liu et al., 2014*). However, the mosses are replaced by the liverworts in the same position when analysing codon-degenerate recoded data. Codon degenerate recoding is used to eliminate synonymous substitutions, which are unconstrained by selection at the protein level and therefore can be subject to high rates

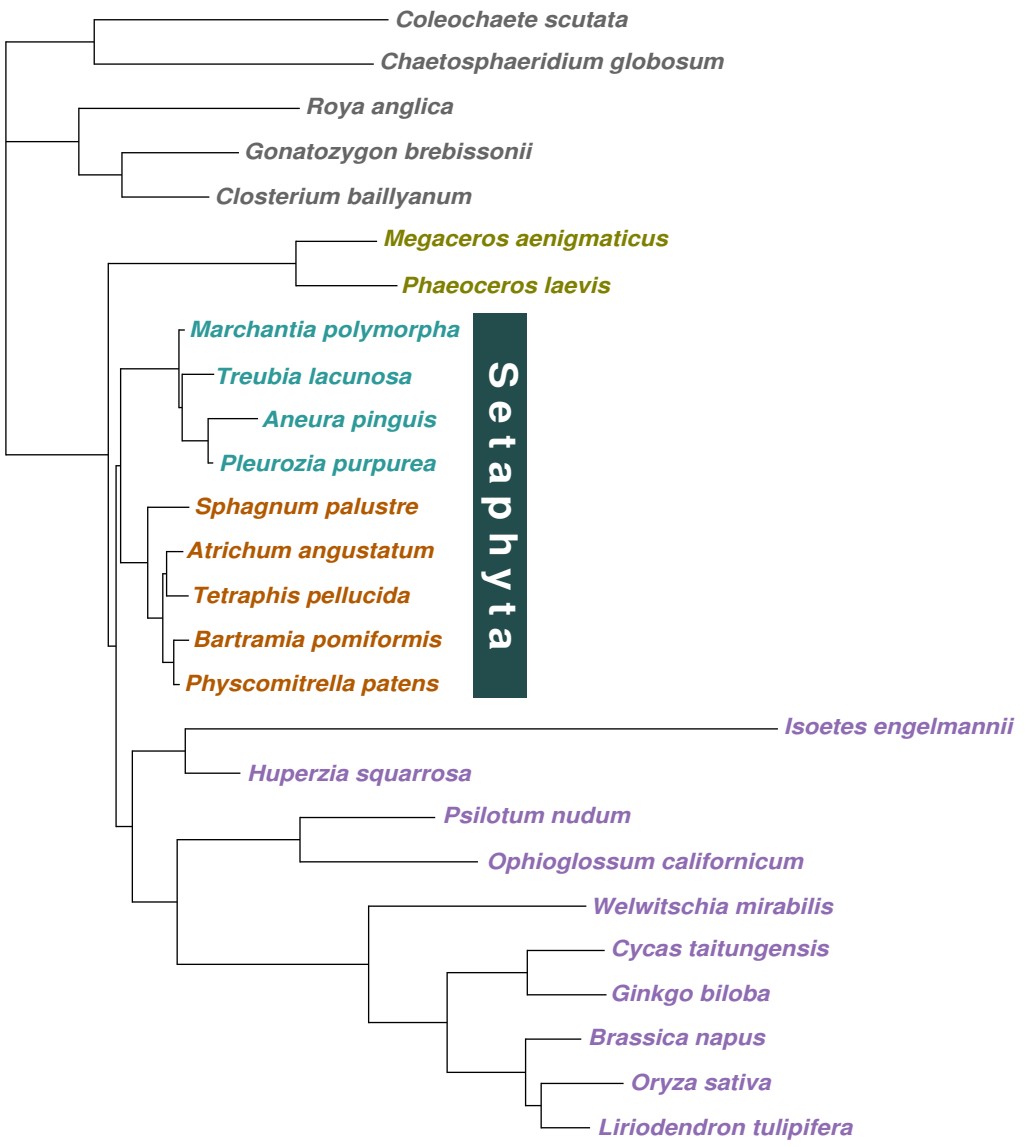

**Figure 2** **Majority-rule consensus tree inferred from the 36 gene, 26 taxon concatenated amino acid data.** Bayesian MCMC with a tree-heterogeneous composition model NDCH2, marginal likelihood -L_h = 142829.112. Additional analysis statistics can be found in the legend of Fig. S7. All branches fully supported (PP = 1.0). Taxa indicated as follows: outgroups, grey; hornworts, olive green; liverworts, cyan blue; mosses, orange; vascular plants, violet.

of substitution, and ultimately saturation and loss of phylogenetic signal. Indeed, as the exclusion of synonymous substitutions is sufficient to eliminate phylogenetic signal that supports mosses as the sister-group to the remaining land plants, these results illustrate that despite being the slowest evolving genomic compartment in plants, phylogenetic inferences from highly divergent mitochondrial genomes are also affected by substitutional saturation due to the effect of high substitution rates. Moreover, this observation implies that caution should be taken when invoking biological explanations (e.g., hybridisation, incomplete
lineage sorting) to explain incongruence among phylogenies when inadequate modeling of the substitution process may represent a simpler explanation.

## Codon degenerate nucleotide data and protein data support Setaphyta under tree-heterogeneous models of substitution

With the nucleotide data, both the tree-homogeneous and NDCH2 tree-heterogeneous models support mosses as the earliest-diverging group in the embryophytes. The likely incorrect placement of mosses as the earliest-diverging group that is recovered with the best-fitting tree-heterogeneous NDCH2 model suggests that homoplasy driven by high nucleotide substitution rates (saturation) may overwhelm the ability of the model to correct for composition bias, hence the need to use codon-degenerate recoded data in combination with tree-heterogeneous models. Indeed, when codon-degenerate recoded data are used, contrasting supported relationships are obtained under tree-homogeneous and tree-heterogeneous composition models. Whereas using a homogeneous model for the analysis of the codon-degenerate data shows liverworts well supported as the sister-group to other embryophytes, the tree-heterogeneous analysis (NDCH2) model places liverworts as the sister-group to the mosses (clade Setaphyta), with maximum support, and hornworts as the sister-group to all other embryophytes, also with maximum branch support. These results demonstrate that the phylogenetic signal contained in non-synonymous sites is also subject to composition biases and that tree-heterogeneous composition models are required to model the data effectively. Contrasting results were also obtained when the amino acid data were analysed with tree-homogeneous and tree-heterogeneous models. The tree inferred under the poorly-fitting ($P = 0.0$) homogeneous model (CV1) resolves liverworts as the sister-group to the remaining land plants (PP = 0.82). By contrast, the Bayesian MCMC run with the highest marginal likelihood under the NDCH2 (fitting) model shows strong support for the Setaphyta clade (PP = 1.0 –mosses plus liverworts) with the hornworts as the earliest-branching lineage of embryophytes (PP = 1.0). *Liu et al. (2014)* observed topological congruence between mitochondrial nucleotide and protein data that placed liverworts as the sister-group to all other embryophytes, but this placement of liverworts received low branch support in different analyses, and thus no firm conclusions regarding these cladogenic events were put foward. Importantly, in that study no tree-heterogeneous analyses of the codon-degenerate data were performed, nor was the protein data analysed with more than two composition vectors on the tree. In contrast, our analyses of the codon-degenerate nucleotide data and amino acid data using a better-fitting tree-heterogeneous model resulted in well supported, congruent topologies, strengthening the argument in favour of the analyses presented here, that show support for the clade Setaphyta.

## The land plant mitochondrial phylogeny is partially congruent with nuclear and chloroplast phylogenies

In contrast to previous analyses of the land plant mitochondrial phylogeny, we show that both nucleotide and amino acid data carry signal that joins mosses and liverworts as sister lineages (clade Setaphyta). This phylogenetic signal is typically obscured due to substitution saturation in the nucleotide data and among-lineage composition bias

in both the nucleotide and amino acid data. In the nucleotide data, phylogenetic signal supporting mosses as the sister-group to the remaining land plants is eliminated when codon-degenerate recoded data is analysed, and instead the liverworts are found as the sister-group to all the remaining embryophytes under tree-homogeneous composition models. However, it is only when a combination of codon-degenerate recoding and a better-fitting tree-heterogeneous composition model is used that the mosses and liverworts appear resolved as sister taxa, therefore suggesting that both substitutional saturation and among-lineage composition heterogeneity are important evolutionary processes to be modeled in the nucleotide data. Similarly, the unsupported placement of liverworts as the earliest-branching lineage is obtained using tree-homogeneous composition models with the amino acid data, but better-fitting tree-heterogeneous composition models again support the mosses plus liverwort clade.

Support for the moss-liverwort sister-group relationship has been found in trees previously inferred from nuclear and chloroplast protein-coding data (e.g., *Nishiyama et al., 2004*; *Cox et al., 2014*; *Puttick et al., 2018*; *Sousa et al., 2019*). The clade can be resolved by mitochondrial data with our analyses, and therefore avoids the necessity of calling upon biological explanations, such as hybridisation, to account for incongruence among the phylogenies of the three plant genomes regarding the placement of mosses and liverworts. However, if the placement of the hornworts as the earliest-branching lineage of embryophytes does indeed reflect the true mitochondrial topology, then it is in conflict with the nuclear and chloroplast data which suggest the bryophytes are likely monophyletic. A biological process involving a rapid divergence of the hornworts from other bryophytes, after the tracheophyte-bryophyte split, and the retention of a copy of the mitochondrion that was lost in all other embryophyte lineages, could be invoked to explain the observed phylogenetic conflict. However, the incongruence of the mitochondrial data could, of course, still be a result of mis-modeling or lack of phylogenetic signal. It is likely that further sampling of mitochondrial genomes from hornworts and other bryophyte lineages may aid resolution of the phylogeny, but such analyses would only be informative if they were in combination with the heterogeneous composition models that have been shown here to be necessary to correctly model the underlying processes of mitochondrial evolution.

## CONCLUSIONS

The main contribution of this study is the demonstration that liverworts are not the sister-group to embryophytes in the land plant mitochondrial phylogeny, unlike earlier analyses of mitochondrial genomes suggested (*Liu et al., 2014*). Instead, strong support is found for the clade Setaphyta, corroborating support for this clade found in nuclear and plastid genomes, and showing that the mitochondrial phylogeny of land plants is not strongly incongruent with the nuclear and plastid phylogenies. Although the best-scoring tree found by analyses of amino acid data places hornworts as sister-group to embryophytes, the monophyly of bryophytes, which is supported by evidence from nuclear and plastid genomes, cannot be strongly rejected. Importantly, this study also shows that modeling

of composition tree-heterogeneity in amino acid data must not be disregarded, even in slower-evolving genomic regions such as plant mitochondria.

### Funding
This study received Portuguese national funds from FCT - Foundation for Science and Technology through project UIDB/04326/2020, and from the operational programmes CRESC Algarve 2020 and COMPETE 2020 through projects EMBRC.PT ALG-01-0145-FEDER-022121 and BIODATA.PT ALG-01-0145-FEDER-022231. This work was also supported through FCT project grant PTDC/BIA-EVF/1499/2014 to Cymon J. Cox. The funders had no role in study design, data collection and analysis, decision to publish, or preparation of the manuscript

### Grant Disclosures
The following grant information was disclosed by the authors:
Foundation for Science and Technology (FCT): UIDB/04326/2020.
CRESC Algarve 2020 and COMPETE 2020: EMBRC.PT ALG-01-0145-FEDER-022121.
BIODATA.PT ALG-01-0145-FEDER-022231.
FCT project grant: PTDC/BIA-EVF/1499/2014.

### Competing Interests
The authors declare there are no competing interests.

### Author Contributions
- Filipe Sousa conceived and designed the experiments, performed the experiments, analyzed the data, prepared figures and/or tables, authored or reviewed drafts of the paper, and approved the final draft.
- Peter Civáň and João Brazão analyzed the data, authored or reviewed drafts of the paper, generated data, and approved the final draft.
- Peter G. Foster conceived and designed the experiments, authored or reviewed drafts of the paper, and approved the final draft.
- Cymon J. Cox conceived and designed the experiments, performed the experiments, analyzed the data, authored or reviewed drafts of the paper, and approved the final draft.

### Data Availability
All the raw data is publicly available at NCBI GenBank: MK720950, MK720948, MK720949.

Curated sequence alignments and tree files are available at Zenodo: Sousa F. et al. (2019). "Data from the article "The mitochondrial phylogeny of land plants shows support for Setaphyta under non-stationary substitution models"". Zenodo. Dataset. DOI: 10.5281/zenodo.3554149.

## Supplemental Information

Supplemental information for this article can be found online at http://dx.doi.org/10.7717/peerj.8995#supplemental-information.

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
