# Peer review of "The mitochondrial phylogeny of land plants shows support for Setaphyta under composition-heterogeneous substitution models"

_PeerJ, doi:10.7717/peerj.8995_

## Round 0.1 · original submission · Major Revisions

Dear Dr. Sousa,

The reviewers think your analyses on the phylogeny of early land plants based on mitochondrial DNA are informative and worth to publish. Nevertheless, they have some concerns that have to be addressed. In particular, a reviewer has a concern about the representativeness of the single genes/data types and the whole data universe of the organisms. For this reason, he suggests permutation analyses to further evaluate the effect of taxon sampling on performance of certain genes in your phylogenetic reconstruction. Moreover, reviewers ask to go deep in discussing several results as for example, which lineage of charophytes share the closest common ancestor with land plants and to discuss briefly some anomalous results (e.g., the recovery of Marchantia, rather than Treubia, as sister to the other liverworts in several analyses, and the apparent paraphyly of the gymnosperms). You will find the complete list of reviewers’ suggestions at the end of this letter.

I encourage you to improve the manuscript according to tips of reviewers. Please, respond point-to-point to the comments of reviewers to speed up the process of revision.

Once again, thank you for submitting your manuscript to PeerJ and we look forward to receiving your revision.

Sincerely,
Gabriele Casazza

Reviewer 1 ·

Basic reporting

Review of “The mitochondrial phylogeny of land plants shows support for Setaphyta under composition heterogeneous substitution models (#43726)”

The phylogenetic relationships among three bryophyte lineages and tracheophytes have received a great amount of attention recently, especially from scientists who obtained nuclear genomic and transcriptomic data. Because some fundamental issues regarding gene orthology and the fit between genes with different evolutionary rates and the scope of phylogenetic have not been seriously addressed in studies that used the nuclear data, it is difficult to take the results from those nuclear gene studies seriously. The organellar genes do not have the orthology issue, but their rates can be quite variable and thus are not all suitable to reconstruct the basal land plant phylogeny. In this regard, the authors of this paper, after obtaining three new green alga mitochondrial genomes, performed more careful analyses before discussing the phylogenetic results. My main concern is on the validity of the theoretical standards that can be used to evaluate appropriateness of certain genes and data types for reconstructing the basal land plant phylogeny. I am worried that even a good fit between a model and the data only indicates that the data is OK with the model, but it does not say anything about representativeness of the particular genes/data types and the ENTIRE data universe of the organisms, which should be the result of organism evolution. In this regard, I suggest that the authors should experiment with permutation analyses of different taxon sampling schemes to further evaluate the effect of taxon sampling on performance of certain genes in reconstructing the basal land plant phylogeny. GenBank does have more data than what they have included in this study.

Experimental design

Review of “The mitochondrial phylogeny of land plants shows support for Setaphyta under composition heterogeneous substitution models (#43726)”

The phylogenetic relationships among three bryophyte lineages and tracheophytes have received a great amount of attention recently, especially from scientists who obtained nuclear genomic and transcriptomic data. Because some fundamental issues regarding gene orthology and the fit between genes with different evolutionary rates and the scope of phylogenetic have not been seriously addressed in studies that used the nuclear data, it is difficult to take the results from those nuclear gene studies seriously. The organellar genes do not have the orthology issue, but their rates can be quite variable and thus are not all suitable to reconstruct the basal land plant phylogeny. In this regard, the authors of this paper, after obtaining three new green alga mitochondrial genomes, performed more careful analyses before discussing the phylogenetic results. My main concern is on the validity of the theoretical standards that can be used to evaluate appropriateness of certain genes and data types for reconstructing the basal land plant phylogeny. I am worried that even a good fit between a model and the data only indicates that the data is OK with the model, but it does not say anything about representativeness of the particular genes/data types and the ENTIRE data universe of the organisms, which should be the result of organism evolution. In this regard, I suggest that the authors should experiment with permutation analyses of different taxon sampling schemes to further evaluate the effect of taxon sampling on performance of certain genes in reconstructing the basal land plant phylogeny. GenBank does have more data than what they have included in this study.

Validity of the findings

Review of “The mitochondrial phylogeny of land plants shows support for Setaphyta under composition heterogeneous substitution models (#43726)”

The phylogenetic relationships among three bryophyte lineages and tracheophytes have received a great amount of attention recently, especially from scientists who obtained nuclear genomic and transcriptomic data. Because some fundamental issues regarding gene orthology and the fit between genes with different evolutionary rates and the scope of phylogenetic have not been seriously addressed in studies that used the nuclear data, it is difficult to take the results from those nuclear gene studies seriously. The organellar genes do not have the orthology issue, but their rates can be quite variable and thus are not all suitable to reconstruct the basal land plant phylogeny. In this regard, the authors of this paper, after obtaining three new green alga mitochondrial genomes, performed more careful analyses before discussing the phylogenetic results. My main concern is on the validity of the theoretical standards that can be used to evaluate appropriateness of certain genes and data types for reconstructing the basal land plant phylogeny. I am worried that even a good fit between a model and the data only indicates that the data is OK with the model, but it does not say anything about representativeness of the particular genes/data types and the ENTIRE data universe of the organisms, which should be the result of organism evolution. In this regard, I suggest that the authors should experiment with permutation analyses of different taxon sampling schemes to further evaluate the effect of taxon sampling on performance of certain genes in reconstructing the basal land plant phylogeny. GenBank does have more data than what they have included in this study.

Additional comments

Review of “The mitochondrial phylogeny of land plants shows support for Setaphyta under composition heterogeneous substitution models (#43726)”

The phylogenetic relationships among three bryophyte lineages and tracheophytes have received a great amount of attention recently, especially from scientists who obtained nuclear genomic and transcriptomic data. Because some fundamental issues regarding gene orthology and the fit between genes with different evolutionary rates and the scope of phylogenetic have not been seriously addressed in studies that used the nuclear data, it is difficult to take the results from those nuclear gene studies seriously. The organellar genes do not have the orthology issue, but their rates can be quite variable and thus are not all suitable to reconstruct the basal land plant phylogeny. In this regard, the authors of this paper, after obtaining three new green alga mitochondrial genomes, performed more careful analyses before discussing the phylogenetic results. My main concern is on the validity of the theoretical standards that can be used to evaluate appropriateness of certain genes and data types for reconstructing the basal land plant phylogeny. I am worried that even a good fit between a model and the data only indicates that the data is OK with the model, but it does not say anything about representativeness of the particular genes/data types and the ENTIRE data universe of the organisms, which should be the result of organism evolution. In this regard, I suggest that the authors should experiment with permutation analyses of different taxon sampling schemes to further evaluate the effect of taxon sampling on performance of certain genes in reconstructing the basal land plant phylogeny. GenBank does have more data than what they have included in this study.

Reviewer 2 ·

Basic reporting

This is a well-written and logically structured article. In general I found no faults in the basic reporting, although I did note a few typos, listed below:

Lines 60-61: "...between a clade uniting mosses and liverworts and other embryophytes...". On first pass I read this as clade of all three: mosses, liverworts, and other embryophytes, but I assume the authors mean the clade contains only mosses and liverworts. I would suggest rephrasing this for clarity, to something like: "...between a clade uniting mosses and liverworts as sister to other embryophytes...".
Lines 97-98: There is no Cox et al. (2018) listed in the references; should this be Cox et al. (2014), or Cox (2018)?
Line 182: "non-monphyly" - missing 'o'.
Line 252: "was a not a good" - additional 'a' should be deleted.
Line 261: "ccm" should be in italics.
Line 264: "other" - should be 'others'.
Line 348: "toplogies" - missing 'o'.
Line 369: "relatively to" - '-ly' should be deleted
Line 391: "if there were" - should be 'they'.
Supporting information: Tree legends should be checked carefully for typos - I noticed "composition" was frequently missing one 'i', while "predictive" frequently had an extra 'i'.

Experimental design

This is a coherent study and in general the methods are presented well. Although I think the sampling is a little too sparse and would have liked to see at least a few more representatives in each bryophyte lineage (discussed below), the authors do a good job of justifying their restricted sampling due to the computational complexity of the analyses.

I do think there is some inconsistency with the discussion (and selection) of taxonomic sampling, and I think the sampling strategy could be better explained. In the abstract and introduction the authors imply that they are interested in six lineages, three each in bryophytes and tracheophytes (Lines 29-30, 44, 47-48), and in the methods state that they had a deliberately restricted sampling to provide a “balanced selection” between the bryophytes and tracheophytes (Lines 107-108). However, the remainder of the paper treats the tracheophytes as a single lineage, and bryophytes as three distinct lineages (Lines 180-181, 291, 299), which suggests that for a "balanced selection" a similar sampling for each of the four lineages under investigation would be required. The authors also stress that the addition of fern taxa within the tracheophytes might reduce the possibility of artefacts relating to long-branch attraction (Lines 370-372), but don’t address this within the bryophyte lineages.
Therefore, I think the paper would benefit from a better explanation of the breakdown of each lineage -- why so few hornworts for example? Why no representatives of the Jungermanniidae or Hypnanae (the most speciose groups of liverworts and mosses, respectively)? Is this simply due to data availability?
I understand that with each additional sample the increase in computational complexity is not linear, but I feel a few more samples to balance each bryophyte lineage with the tracheophytes would have been useful, without the analyses becoming intractable. I'm not suggesting the authors revisit their alignments and analyses at this point, but I do feel a more thorough explanation would improve clarity.

Some additional minor queries on the methods are listed below:

Line 156: What was the cause of missing genes (incomplete genomic coverage, genes transferred to the nucleus in some lineages, etc)?

Lines 181-183: Why were aberrant genes considered unsuitable (it seems there is an assumption of the monophyly of each lineage, but I'm not sure that sources supporting this were explicitly cited previously?)

Line 223: Will the ML trees be included among the files on Zenodo? It would be good to have them available somewhere, although including them in the supporting information file would be preferred.

Validity of the findings

No comment.

Additional comments

I enjoyed reading this article on the mitochondrial phylogeny of land plants, which adds to a growing body of work using genomic data to attempt to resolve land plant relationships, and ties in well with other recent studies based on nuclear and plastid genomic data sets. This study is particularly useful in that it adds to the relatively limited literature on mitochondrial phylogenomics of land plants (compared to the more abundant nuclear and plastid studies; although please see the very recent article in ‘early view’ in AJB as an additional study of land plant relationships using plastid and mitochondrial data sets: https://doi.org/10.1002/ajb2.1397), as well as further investigating the techniques appropriate for analysing these sorts of data, particularly with reference to computational limitations which will continue to be an issue for phylogenomic analyses as richer data sets become available.

While relationships within lineages is not the focus of the paper, I think it would be worthwhile for the authors to briefly discuss the anomalous results inferred in some of their trees, namely the recovery of Marchantia, rather than Treubia, as sister to the other liverworts in several analyses, and the apparent paraphyly of the gymnosperms (both situations shown in Figure 2 with maximum support, but contrary to a growing body of evidence -- e.g. Forrest et al. 2006 - The Bryologist 109: 303-334; Liu et al. 2014; Wickett et al. 2014). If this apparent mis-inference within lineages is the result of saturation effects in long branches, then it somewhat undermines the argument that these artefacts can be better dealt with by more complex modelling in restricted sampling than by increasing taxonomic coverage to break up long branches. As was suggested with ferns in the tracheophytes here (Lines 370-372), additional sampling within each bryophyte clade may have helped reduce these effects (e.g. see Pollock et al. 2002 - Syst Biol 51: 664–671; Hedtke et al. 2006 - Syst Biol 55: 522–529).

Some additional comments not addressed above:
Lines 35, 105-107, 372-373:
Zygnematales were represented in the sampling of Turmel et al. (2013). Additionally, Bell et al. (https://doi.org/10.1002/ajb2.1397) sampled several ferns and Zygnematales in their mitochondrial analyses.

Lines 99-101:
Or in conflict with different analyses from within the same genomic compartment! Most of the cited studies did recover conflicting topologies for the same genomic compartment, depending on how they treated the data.

Lines 384-387:
The short branches linking each subsequent split among the four major lineages here, relative to the long branch between algae and embryophytes (Fig. 2 and supporting information) may also be a factor here -- is this something the authors have considered? Is there scope to develop more suitable models to tackle the saturation effects which may mislead inference due to this extreme long branch situation?

Reviewer 3 ·

Basic reporting

In this study, the authors using the mitochondrial (mt) genome data to test the controversial question about the deep phylogenetic relationships of early land plants. They newly sequenced mt genomes from three algae, including two from Zygnematales, the putative closest outgroup of land plants. They also include ferns that were not sampled in the previous study by Liu et al. 2014. With a dataset of 21 land plants and 5 algal outgroups, they performed phylogenetic analyses of nucleotide, amino acid, and codon-degenerate nucleotide data, under the tree-homogeneous or tree-heterogeneous composition models, respectively. Their results indicated that under the tree-homogeneous model the three bryophyte lineages are paraphyletic, while under the tree-heterogeneous model, the three bryophyte lineages are paraphyletic or mosses and liverworts form a clade, i.e., Setaphyta is recovered. However, a monophyletic relationship of bryophytes was never recovered by the mt genome data. The research carried out intensive analyses to test the phylogeny of early land plants using the mt genome data, which are informative and worth to publish, however some issues as listed below should be addressed.

1. Although the authors explained due to the computational restriction, only a small size of samples (21 land plants) were included, they should still discuss about this limitation, considering more than 100 land plant mt genomes are available in GenBank. The drawbacks of employing reduced taxon sampling have been long discussed.

2. As I understand, the purpose of degenerating codons is to reduce the effects of base composition. It is amazing the data still cannot pass the homogeneity test. The logic of using a heterogeneous model on a heterogeneous “treated” dataset seems odd, please explain.

3. Both the codon-degenerate data and amino acid dataset cannot pass the homogeneity test, that means some lineages are biased on GC composition, but the authors did not point out which lineages stand out on GC compositions, i.e., what exactly caused the conflict between the two (hetero- and homo-) models.

4. When the tree-heterogeneous composition NDCH2 model was used to model the concatenated protein data, the two separate runs did not converge on the same tree topology, what is the alternative topology? Does it mean the sampling of the parameters is not sufficient? The authors should consider run the MCMC for more generations.

5. RNA editing is common in plant mt genomes, some lineage such as lycophytes host >2,000 editing sites. Dose RNA editing impact phylogenetic analysis? This point should be discussed.

Experimental design

The new data the authors generated in this study were from three algae, including two Zygnematales, but they did not discuss the question about which lineage of charophytes share the closest common ancestor with land plants. From the trees presented in the manuscript, it is not clear if this question is resolved, as the trees were not rooted. The data the authors generated and the question they focused seem not fit, the authors should explain.

Validity of the findings

no comment

Additional comments

no comment

---

## Round 0.2 · Minor Revisions

Dear Dr. Sousa,

the reviewers think your manuscript was strongly improved. Contrary to a previous study (Liu et al. 2014), you find that Setaphyta and not liverworts are the sister-group to embryophytes in the land plant mitochondrial phylogeny. This is the central point of your manuscript. For this reason, a reviewer still asks you to better explain and/or sustain why you are confident in your result, particularly in relation to the previous work.
Once again, thank you for submitting your manuscript to PeerJ and we look forward to receiving your revision.

Sincerely,
Gabriele Casazza

Reviewer 2 ·

Basic reporting

No further comment.

Experimental design

No further comment.

Validity of the findings

No further comment.

Additional comments

The authors have addressed all concerns raised by the reviewers adequately, and I think the updated text will be a valuable addition to the literature on this topic.

Reviewer 3 ·

Basic reporting

The authors’ response is appreciated, but with regards to the argument on the sparse sampling in this study, the authors’ explanation is still not convincing. The study by Liu et al. (2014) focused on a same question as this study, and they took similar analytical strategies. Actually this study is more or less a reduced set of that study’s sampling (ca. 20 vs 60). The main difference is this study included three newly sequenced algae, and two recent published ferns. But as the authors pointed out adding branches in tracheophytes should not impact the resolution of relationships among major clades.

Since Liu et al. (2014) did not recover a Setaphyta with a larger dataset, what makes the current study recovered it? In other words, which result is more reliable and should be accepted? Isn’t it worth to test whether removing the newly added algal outgroups will lead to a constant result, or including the new algal outgroups in Liu et al. (2014)’s dataset will result in a Setaphyta as in this study? I worry about the consistency and robustness of the novel result (i.e. a Setaphyta) found in this study.

Experimental design

NA

Validity of the findings

NA

Additional comments

NA

---

## Round 0.3 · accepted · Accept

Dear Dr. Sousa,
I carefully read your answer to the reviewer and in the light of your answer I am very pleased to say that your paper "The mitochondrial phylogeny of land plants shows support for Setaphyta under composition-heterogeneous substitution models" is accepted for publication in the PeerJ. Congratulations!

Thank you for submitting your work to PeerJ.

Sincerely,

Gabriele Casazza